# STATES OF LLM-GENERATED TEXTS AND PHASE TRANSITIONS BETWEEN THEM

**Nikolay Mikhaylovskiy**
NTR Labs, Moscow, Russia
and Higher IT School
of Tomsk State University,
Tomsk, Russia
`nickm@ntr.ai`

## ABSTRACT

It is known for some time that autocorrelations of words in human-written texts decay according to a power law. Recent works have also shown that the autocorrelations decay in texts generated by LLMs is qualitatively different from that in literary texts. Solid-state physics tie the autocorrelations decay laws to the states of matter. In this work, we empirically demonstrate that, depending on the temperature parameter, LLMs can generate text that can be classified as solid, critical state, or gas.

## 1 INTRODUCTION

Although not long ago probabilistic autoregressive language models were just models that assign probabilities to sequences of words (Bahl et al., 1983), now they are the cornerstone of any task in computational linguistics by prompting (Sanh et al., 2022) or fine-tuning (Radford et al., 2018). Such models being successfully commercialized, the number of practical applications of these models is rapidly growing, as is the number of papers considering various aspects of the use of probabilistic autoregressive language models. It is all the more surprising that the statistical properties of the output sequences produced by such models have been relatively little studied.

We aim to fill this gap somewhat and empirically demonstrate that, depending on the temperature parameter, LLMs can generate text that can be classified as solid (periodic phase), critical state (that has autocorrelations decay according to the power law), or gas (amorphous phase) from the point of view of autocorrelation analysis.

Our main contributions are the following:

1. We clearly identify three phases of LLM-generated texts - periodic, critical and amorphous
2. We show through computational experiments that for LLM-generated texts, there is a phase transition from ordered to amorphous state at about the same temperatures between 0.7 and 1, for different LLMs
3. We show that for amorphous state, long-range autocorrelations decay follows the exponential law independently from the generation temperature, for different LLMs
4. We show that for temperatures between 0.7 and 1 autocorrelations exhibit power law decay on medium distances of up to 2000 words, implying isles of connectivity of these sizes.

We go on to introduce the key concepts.

### 1.1 AUTOREGRESSIVE PROBABILISTIC LANGUAGE MODELS

Probabilistic language models consider sequences

$$t_{1:m} = \{t_1, t_2, \ldots, t_m\} \tag{1}$$

of tokens from the lexicon $L$. An autoregressive language model estimates the probability of such a sequence

$$P(t_{1:m}) = P(t_1)P(t_2|t_1)P(t_3|t_{1:2})\dots P(t_m|t_{1:m-1})$$
$$= \prod_{k=1}^{m} P(t_k|t_{1:k-1}) \tag{2}$$

using the chain rule. Many models introduce the Markov (1913) assumption that the probability of a token depends on the previous $n-1$ tokens only, thus approximating (3) with a truncated version

$$P(t_{1:m}) \approx \prod_{k=1}^{m} P(t_k|t_{k-n+1:k-1}) \tag{3}$$

## 1.2   TEXT GENERATION WITH A LANGUAGE MODEL

Given an input text as a context, the goal of open-ended generation is to produce a coherent continuation of the text (Holtzman et al., 2020). More formally, given a sequence of $m$ tokens $t_1 \dots t_m$ as context, the objective is to generate the next $n$ continuation tokens, resulting in the completed sequence $t_1 \dots t_{m+n}$. This is achieved through the use of the left-to-right text probability decomposition (2), which is used to generate the sequence one token at a time, using a particular decoding strategy.

A common approach to text generation is to shape a probability distribution through temperature (Ackley et al., 1985). Given the logits $u_{1:|V|}$ and temperature $T$, the softmax is re-estimated as

$$p(t = V_l|t_{1:i-1}) = \frac{\exp(u_l/T)}{\sum_{l'} \exp(u_{l'}/T)} \tag{4}$$

Setting $T \in [0, 1)$ skews the distribution towards high-probability events, and, similarly, $T \in (1, \infty)$ skews the distribution towards low-probability events.

## 1.3   COMPUTING AUTOCORRELATIONS USING DISTRIBUTIONAL SEMANTICS

Suppose that we have a sequence of $N$ vectors $V_i \in R^d, i \in [1, N]$. The autocorrelation function $C(\tau)$ is the average similarity between the vectors as a function of the lag $\tau = i - j$ between them. The simplest metric of vector similarity is the cosine similarity

$$d(V_i, V_j) = \cos(V_i, V_j) = \frac{(V_i \cdot V_j)}{||V_i||||V_j||}, \tag{5}$$

where $\cdot$ is a dot product between two vectors and $||x||$ is an Euclidean norm of a vector. Thus,

$$C(\tau) = \frac{1}{N - \tau} \sum_{i=1}^{N-\tau} \frac{V_i \cdot V_{i+\tau}}{||V_i||||V_{i+\tau}||} \tag{6}$$

A distributional semantic (Harris, 1954) such as GloVe (Pennington et al., 2014) assigns a vector to each word or context in a text. Thus, a text is transformed into a sequence of vectors, and we can calculate an autocorrelation function for the text.

## 1.4   PHASE TRANSITIONS

A physical phase of a system refers to a (typically equilibrium) state with unique macroscopic properties. These phases possess certain stability regions within the parameter space. The properties of the state change at the boundaries of these regions, where phase transition occurs.

Ehrenfest (1933) defined a phase transition as a discontinuity in the n-th order derivative of the free energy with respect to any argument of the free energy. Modern physics extends the notion of phases and applies it to various situations and beyond the notion of free energy. In particular, a first-order phase transition exhibits a discontinuity in the first-order derivative, whereas a second-order phase transition is continuous in its first derivative but shows a discontinuous or divergent behavior in its second derivatives (Papon et al., 2007).

> and in the act of devouring a man,
> and in the act of devouring a whale,
> and in the act of devouring a ship,
> and in the act of devouring a man,
> and in the act of devouring a whale,
> and in the act of devouring a ship,
> and in the act of devouring a man,
> and in the act of devouring a whale,
> and in the act of devouring a ship,

Figure 1: Degenerative Text Generated by Qwen at t=0.1, shift 11904, seed 1

> My companjour's father held in trust as old, well wizdom; being like ye 'ear say my first—(thro' not only bubbynee I tell) old Cronicle for it were not. No man can give 'fitt or show o more clear-aifv intelligence—that will prove well at last I sase now believe it all me blawms—be it or'tt how.

Figure 2: Nonsense Text Generated by Phi at t=2.8, shift 539, seed 1

## 2 PHASES IN LLM-GENERATED TEXTS

### 2.1 PRIOR RESEARCH

Power-law autocorrelations decay in human-written texts was studied in a number of works (Li, 1989; Schenkel et al., 1993; Ebeling & Pöschel, 1994; Ebeling & Neiman, 1995; Kokol et al., 1999; Pavlov et al., 2001; Montemurro & Pury, 2002; Alvarez-Lacalle et al., 2006; Manin, 2008; Gillet & Ausloos, 2008; Corral et al., 2009; Altmann et al., 2012; Amit et al., 1994; Tanaka-Ishii & Bunde, 2016; Lin & Tegmark, 2017; Takahashi & Tanaka-Ishii, 2017; Shen, 2019; Takahashi & Tanaka-Ishii, 2019; Sainburg et al., 2019; Mikhaylovskiy & Churilov, 2023; Nakaishi et al., 2024).

Significantly less works are devoted to the analysis of statistical quantities in texts generated by language models. Generated texts have been studied by (Takahashi & Tanaka-Ishii, 2017; Shen, 2019; Takahashi & Tanaka-Ishii, 2019; Lippi et al., 2019; Mikhaylovskiy & Churilov, 2023), who conjectured that power-law decays in autocorrelations are model-dependent.

Nakaishi et al. (2024) and Bahamondes (2023) independently pioneered the application of the phase transition apparatus to LLM-generated texts. They both used only GPT-2, which has no practical interest by now. Bahamondes (2023) studied a setup similar to ours but came up with a significantly different phase transition temperature ( 0.1 instead of  0.8). We strongly believe that our results are more accurate. Nakaishi et al. (2024) used part-of-speech correlations that are hardly applicable in the high-temperature area (cf. Figure 3).

### 2.2 TEXT GENERATION SETUP

We use two compact LLMs: Qwen2.5-1.5B (Qwen et al., 2025) and Phi-3-Mini-128K-Instruct (Abdin et al., 2024) to generate texts using the HuggingFace transformer library (Wolf et al., 2020) with temperature sampling (Ackley et al., 1985), similarly to Nakaishi et al. (2024). We further call these models Qwen and Phi for brevity. It is worth noting that from the practical viewpoint these models belong to different classes, as Phi is instruction-tuned, and Qwen is just pretrained.

We start all sequences with the starting passage of "MOBY-DICK; or, THE WHALE." By Herman Melville and iterate the random seed for reproducibility. We force and repeat generation until the text length of 10000 is achieved. We do not use top-k (Fan et al., 2018) or top-p (Holtzman et al., 2020) sampling strategies. Texts are tokenized by a default tokenizer for each model. We sample 10 sequences at each temperature from 0.1 to 2.5 with a step of 0.3. Sampling with each model took less than a day using a single NVIDIA A100.

> I don not what and what is he tal(ed in). The "Bill (s)" said he knew more , the ( 'he knew . Here .....there is so not ... 's, for example... A lot ... In The News-'Vacatio'(as 'He Did S.Y.(se) in Ital:) I(he) sited it on a '(l'he ') page in U S. I know.. , so in 97 :, ,: and , the u..? A l i N o s ,

Figure 3: Gibberish Text Generated by Phi at t=2.8, shift 9794, seed 1

> But I have a little more to say. What I want you to understand is that every one of us has a right to decide when we're old.
> When I started out, a man might think that was a silly idea. I didn't think so.

Figure 4: Text Generated by Phi at t=1.0, shift 9816, seed 1

## 2.3 Computing Autocorrelations

We use pretrained multilingual GloVe vector embeddings by Ferreira et al. (2016) similarly to Mikhaylovskiy & Churilov (2023). Unlike them, we do not filter out any words. We center the vector system by subtracting the average of vectors over the whole text. If a word produced by a tokenizer does not have any pretrained vector corresponding, we assign an all-zeroes vector to the token after centering and assume that in this case all correlations are equal to zero. After that we can compute the autocorrelation function following Section 1.3.

## 2.4 Empirical Phase Observations

It is known for some time (Kulikov et al., 2019; Holtzman et al., 2018; Fan et al., 2018) that at low temperatures LLMs tend to generate degenerate, repetitive text (Figure 1). At high temperatures, the text progresses from nonsense to gibberish in the course of generation (Figure 2, Figure 3). At moderate temperatures, the text is often locally consistent, although it lacks the global consistency and richness characteristic of human-written texts (Figure 4). At such temperatures during long generation we can sometimes witness transitions from random or gibberish text back to human-like one and then decay back to random (See Appendix 1 [1]). From these empirical observations, we can conjecture that the LLM-generated texts can have phase transitions during the generation process (exhibit domains of different phases in space) and as the temperature varies.

## 2.5 Periodic Phase and Its Phase Transition

At low temperatures, the generated text degenerates and becomes periodic (see Figure 1). The periods vary with the seed, temperature, and model. To quantify this, we compute autocorrelations for the distances from 1 to 100 words and plot the autocorrelation function. To further analyze the periodic nature of the autocorrelations, we perform discrete FFT of the autocorrelation function (see Appendix 2 [2] for the extended image set). A typical example of a periodic autocorrelation function is presented in Figure 5, and a typical example of aperiodic autocorrelation function is presented on Figure 6

The difference is obvious on both autocorrelation and FFT graphs. To further quantify the transition between periodic and non-periodic structures, we plot the maximum absolute value of the Fourier transform of the normalized autocorrelation function beyond the first coefficient against the temperature (Figures 7 and 8). It is clear from the figures that the phase transition for both models happens at $t$ around $0.8$. At $t = 1$ the text is non-periodic. The abrupt change is characteristic of a phase transition, as opposed to a gradual change (cf. Section 1.4).

## 2.6 Amorphous Phase

At high temperatures the generated texts are random (see Figure 3 for example). This entails an exponential decay of autocorrelations. At medium temperatures the generated texts are legible and

---

[1]Appendices are available as supplementary material by this link
[2]Appendices are available as supplementary material by this link

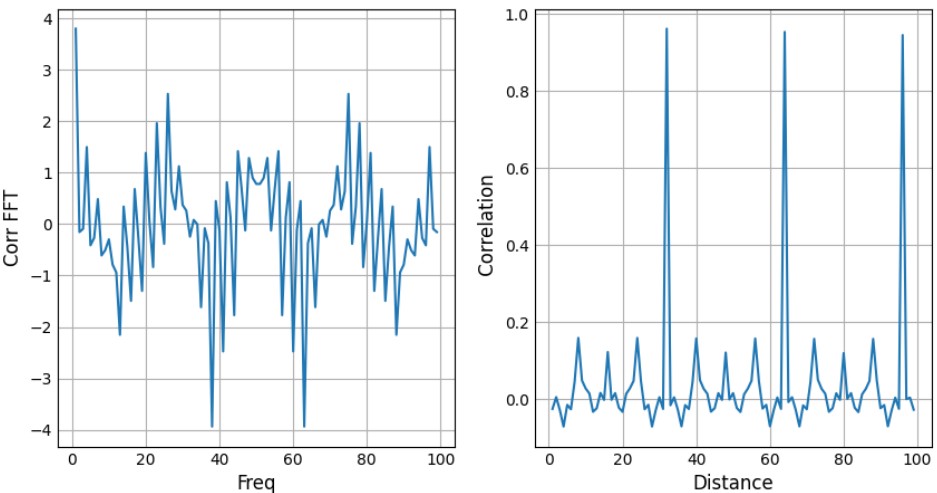

Figure 5: Autocorrelation Function of the Text Generated by Phi at $t = 0.4$ and $seed = 2$ and Its FFT

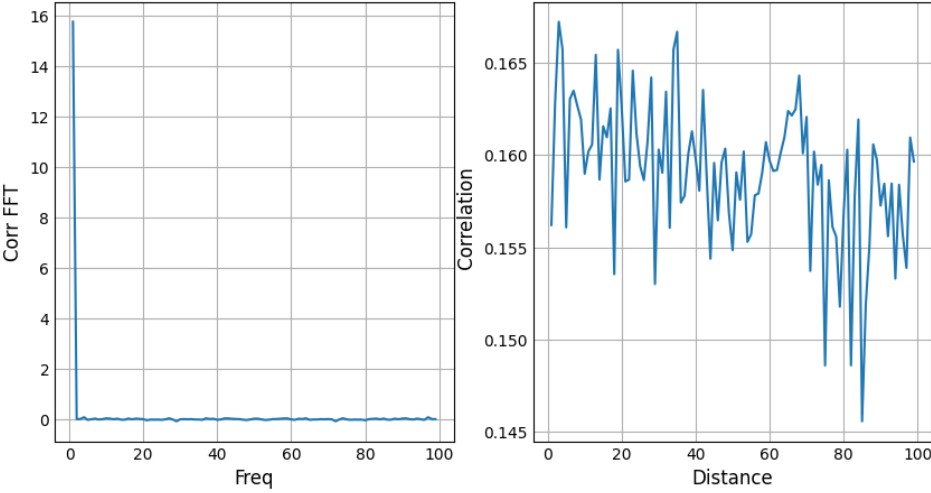

Figure 6: Autocorrelation Function of the Text Generated by Phi at $t = 2.8$ and $seed = 5$. and Its FFT

often exhibit power law autocorrelations decay. We want to study the transition between these two phases.

To quantify this, we compute autocorrelations for selected distances from 1 to thousands of words (see though the further discussion), and plot the autocorrelation function in log and linear coordinates. Periodic texts do not make much sense in log coordinates because there are typically a lot of negative correlations, so we only consider temperatures that are greater than $0.7$. The autocorrelation functions of many texts appear messy (Figure 9), but for other texts one can definitely spot either power law (Figure 10) or exponential autocorrelations decay (Figure 11). See Appendix 3 [3] for the extended image set.

Mikhaylovskiy (2023) suggested GAPELMAPER (GloVe Autocorrelations Power/ Exponential Law Mean Absolute Percentage Error Ratio) metric to distinguish texts with exponential and power law autocorrelations decay and determine whether the text has a hierarchical structure. Application of this metric to the texts up to long-range correlations shows that most LLM-generated texts have

---

[3]Appendices are available as supplementary material by this link

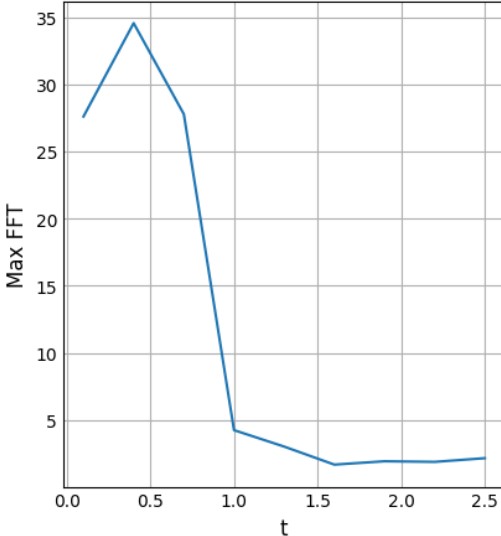

Figure 7: Transition from Periodic to Amorphous Phase in Phi-generated Texts

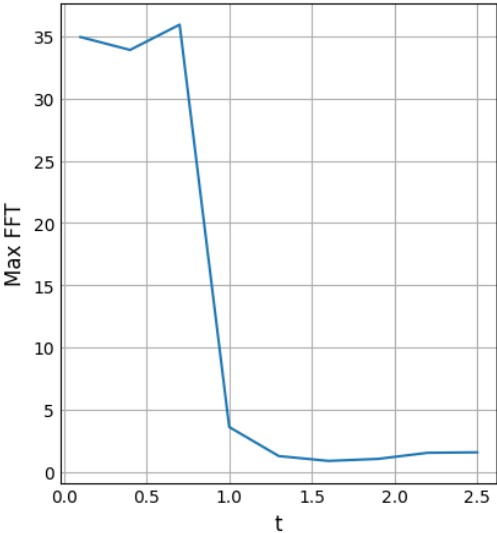

Figure 8: Transition from Periodic to Amorphous Phase in Qwen-generated Texts

exponential autocorrelations decay and thus no inherent hierarchical structure. For example, if we compute autocorrelations up to a distance of 6000 words in Qwen-generated texts we will observe that for no temperature there is a certain power law decay (see Figure 12), that is, GAPELMAPER is rarely if ever less than 1 (GAPELMAPER cannot be computed for most texts generated with $t \in \{0.1, 0.4, 2.8\}$ because of negative autocorrelations).

On the other hand, if we limit the autocorrelations distance to, say, 600 words, we will observe that at $t = 0.7$ GAPELMAPER is reliably less than 1, and wiggles around 1 at higher temperatures (see Figure 13). Only at autocorrelation lengths around 3000 the situation changes qualitatively. This means that the power correlations exist in shorter (under 2000 words) chunks of the generated texts at temperatures of 0.7 to 1.0, but the correlations are lost at longer distances. See Appendix 4 [4] for more data on this.

---

[4] Appendices are available as supplementary material by this link

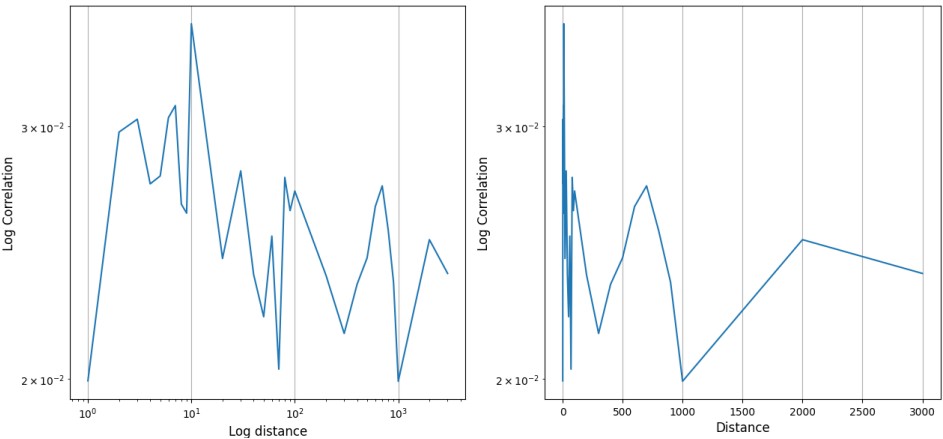

Figure 9: Autocorrelation Function for the Text Generated by Phi at $t = 1.0$ and $seed = 7$

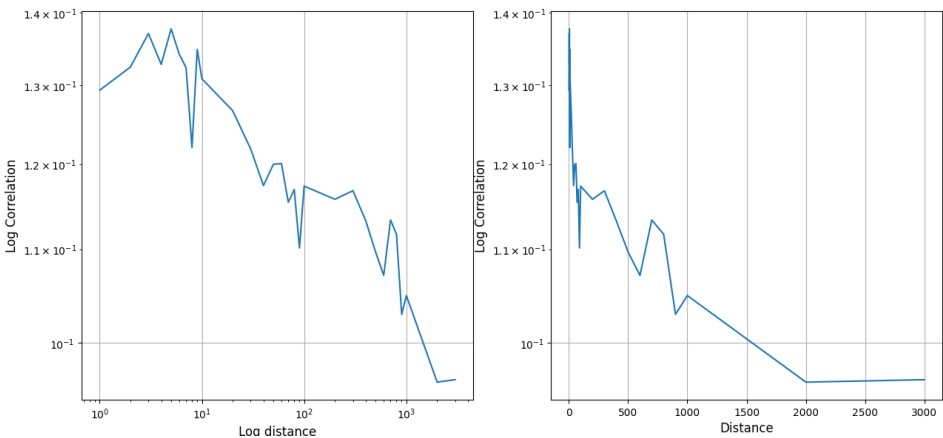

Figure 10: Autocorrelation Function for the Text Generated by Phi at $t = 1.9$ and $seed = 1$

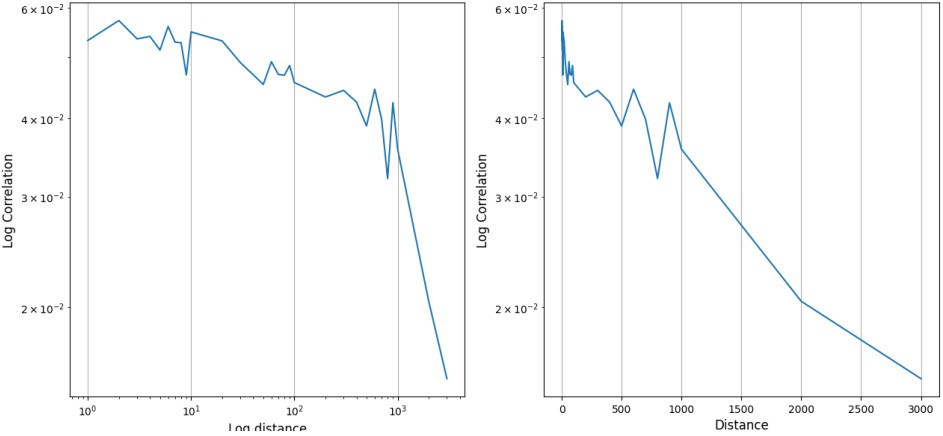

Figure 11: Autocorrelation Function for the Text Generated by Phi at $t = 2.8$ and $seed = 8$

## 3 DISCUSSION AND FUTURE WORK

The use of the phase transition apparatus to study LLM-generated texts promises a deeper under-standing of both human- and LLM-generated texts, as well as the mechanisms of inner workings

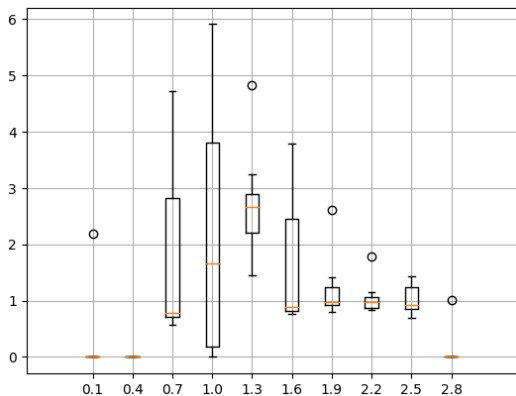

Figure 12: GAPELMAPER Metric for Qwen-Generated Texts, Autocorrelation Distances 1 to 6000 Words

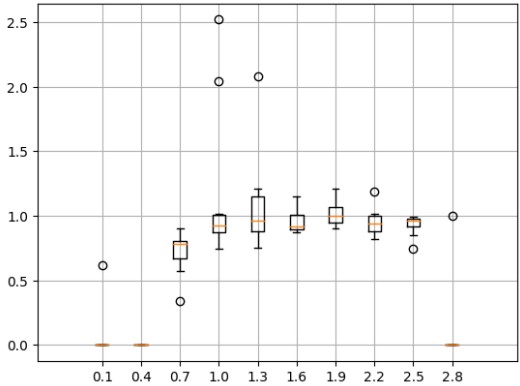

Figure 13: GAPELMAPER Metric for Qwen-Generated Texts, Autocorrelation Distances 1 to 600 Words

of LLMs. We have clearly identified three phases of LLM-generated texts - periodic, critical, and amorphous. Our research confirms the existence of a phase transition in LLM-generated texts at a temperature of about $0.8$, which largely confirms the results of Nakaishi et al. (2024). We have also shown for the first time that for the amorphous state, the long-range autocorrelations decay follows the exponential law independently of the generation temperature, for different LLMs, which was previously conjectured by Mikhaylovskiy & Churilov (2023).

We have shown that for temperatures between $0.7$ and $1$ autocorrelations exhibit power law decay on medium distances of up to 2000 words. This implies islets of connectivity of these sizes, but the fine-grained structure of this sort remains unexplored. This is a topic for the future research.

Our results with different LLMs allow us to conjecture that transformer-based LLMs belong to the same universality class, as described in statistical physics. Do other models of different architecture belong to this class, say, state-space-based or diffusion-based, remains unexplored and is a topic of future research. Additional research is also needed to study the influence of LLM size on the universality class.

ACKNOWLEDGEMENTS

The author thanks Dmitry Manin and Kai Nakaishi for discussing earlier versions of this work.

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
