# OpenReview forum: "States of LLM-generated Texts and Phase Transitions between them"
_mathai.club/MathAI/2025/Conference — MathAI 2025 Oral_

### Official Review · Reviewer_A2C7 · 2025-02-26
**Proper base for future research**

**Rating:** 7
**Confidence:** 4

**Review:**

The authors identify three phases of LLM-generated texts: periodic, critical, and amorphous. They then explore the impact of temperature on state transitions, outlining how this behavior resembles a phase transition.

**Strengths**
- The paper presents a connection between physical concepts and LLM-generated text.
- The identification of phase transitions is valuable for many prompting applications, which lack such approach.
- The results align well with previous research by Mikhaylovskiy and support the initial phase transition hypothesis.
- The paper is well-structured, free of typos, and features clear, self-explanatory figures.

**Downsides**
- The prior research section is relatively short and could provide more context.
- The absence of an appendix limits the reliability of certain results. For instance, only a few examples of the autocorrelation function are presented without accompanying statistical analysis. Additionally, extended GAPELMAPER data is missing, reducing insight into long-range correlations.
- The authors do not explicitly differentiate their findings from previous research on the same topic. Without statistical validation through multiple seeds and initial texts, it is unclear whether the observed phase transition consistently occurs within the same temperature range. No insights were added concerning the experimental setup flaws in previous works.

**Conclusions**
Overall, the results are promising. However, incorporating statistical validation and testing across multiple language models would strengthen the findings on phase transition ranges. This work serves as a strong foundation for further research into the impact of different model architectures, sizes, and other factors on critical temperatures. Additionally, deeper analysis of the underlying reasons for phase transitions and variations across models would enhance the paper’s contribution.

**Remarks**
- Line 067: Possible typo in "degeneretive."
- Citation hyperlinks are missing and should be included.
- It would be beneficial to add data points to graphs (e.g., Figures 7-8) and provide statistical results accounting for different seeds.

---

### Official Review · Reviewer_gQ3V · 2025-02-27
**Review of paper 42**

**Rating:** 6
**Confidence:** 3

**Review:**

**Broad Review**

This paper explores the statistical properties of texts generated by LLMs. It examines how the autocorrelation decay laws of these texts can be linked to the states of matter (solid, critical state, and gas) and discusses the phase transitions between these states based on the temperature parameter used during text generation.

**Strengths**

•	The demonstration of phase transitions at specific temperatures provides new insights into the behavior of LLMs.

•	The application of concepts from solid-state physics to the study of text generation bridges disciplines and offers a fresh perspective on the inner workings of LLMs.

•	This paper does a good job by identifying power law decay at medium distances (up to 2000 words) in LLM-generated texts, especially at temperatures between 0.7 and 1. This is a significant finding because it suggests that there are coherent, connected segments within the generated text at these temperatures

**Weakness**

•	The study focuses on specific LLMs (Qwen and Phi). Authors can strengthen their conclusions by including a broader range of models to determine if the observed phenomena are universal across different LLMs. Different models might have different training data and hyperparameters which can lead to variations in text generation and phase behavior.

•	Citations’ hyperlinking has not been done.

•	Defining the acronym ' GAPELMAPER’ the first time they appear in the paper would enhance reader comprehension and ensure clarity for those unfamiliar with the term.

•	Under Section 1.4, "varies" should be "various".

•	Maintain uniform formatting for all references in the text and the reference list. Example: in-text citations that involve multiple authors are typically simplified using "et al." after the first author's last name. This helps maintain readability and conciseness. So instead of writing "Schenkel A., Zhang J., Zhang Y.-C., 1993" under section 2.1, the authors should use "Schenkel et al., 1993". This indicates that Schenkel is the first author, and there are additional authors as well.

**Conclusion**

Overall, the paper offers a well-supported investigation into the phase transitions in LLM-generated texts. However, it could benefit from broader model consideration. Additionally, the paper's formatting is currently rough and necessitates substantial refinement to be suitable for formal submission.

---

### Official Review · Reviewer_JJrM · 2025-02-27
**Review of paper 42 (Updated)**

**Rating:** 7
**Confidence:** 3

**Review:**

The paper describes the statistical properties of LLM generated texts with different temperatures that might be useful for the AI-generated texts detection. Here is the strengths and weaknesses of the paper.

**Strengths**

- the proposed approach of analyzing the LLM generated text is novel and can be used in practical tasks such as AI-generated texts detection;

- the paper compares the actions inside LLMs with physical world, what can be interesting finding for further investigation.

**Weaknesses**

- the citations hyperlinks are not working;
- the figures 1-3 appeared at the first section, however they was mentioned at the second part of section 2, which is looking not good for readability;
- there was mentions of Appendix 1-4 but no appendix section were included in the paper;
- the study describes the outputs of the Qwen and Phi models, but it might be interesting to include the outputs of wider range of models (including LLaMA family, etc).

The content of paper is novel, its results are useful for further research and practical implementation, but formatting of the paper is not complete. Fixing this issues will make this paper suitable for publishing.

---

### Decision · Program_Chairs · 2025-03-08

**Decision:**

Accept (Oral)

**Comment:**

Your article has been accepted and you can make a presentation on the article. All articles will be sorted by rating and within the available conference places one author from each article will be invited. If there are not enough places, then you will either have the opportunity to present remotely or come at your own expense!